

# The sensitivity of PM$_{2.5}$ acidity to meteorological parameters and chemical composition changes: 10-year records from six Canadian monitoring sites

Ye Tao[1], Jennifer G. Murphy[2]

[1]Department of Physical and Environmental Sciences, University of Toronto Scarborough, Toronto, ON, M1C 1A4, Canada

[2]Department of Chemistry, University of Toronto, Toronto, ON, M5S 3H6, Canada

*Corresponding to*: Jennifer G. Murphy (jmurphy@chem.utoronto.ca)

**Abstract.** Aerosol pH is difficult to measure directly but can be calculated if the chemical composition is known with sufficient accuracy and precision to calculate the aerosol water content and the H$^+$ concentration through ion balance. In practical terms, simultaneous measurements of at least one semi-volatile constitute, e.g. NH$_3$ or HNO$_3$, are required to provide a constraint on the calculation of pH. Long-term records of aerosol pH are scarce due to the limited monitoring of NH$_3$ in conjunction with PM$_{2.5}$. In this study, 10-year (2007-2016) records of pH of PM$_{2.5}$ at six eastern Canadian sites were calculated using the E-AIM II model with the input of gaseous NH$_3$, gaseous HNO$_3$ and major water-soluble inorganic ions in PM$_{2.5}$ provided by Canada's National Air Pollution Surveillance (NAPS) Program. Clear seasonal cycles of aerosol pH were found with lower pH (~2) in summer and higher pH (~3) in winter consistently across all six sites, while the day-to-day variations of aerosol pH were higher in winter compared to summer. Tests of the sensitivity of aerosol pH to meteorological parameters demonstrate that the changes in ambient temperature largely drive the seasonal cycle of aerosol pH. The sensitivity of pH to chemical composition shows that pH has different responses to the changes in chemical composition in different seasons. During summertime, aerosol pH was mainly determined by temperature with limited impact from changes in NH$_x$ or sulfate concentrations. However, in wintertime, both meteorological parameters and chemical composition contribute to the variations in aerosol pH, resulting in the larger variation during wintertime. This study reveals that the sensitivity of aerosol pH to chemical composition is distinctly different under different meteorological conditions and needs to be carefully examined for any particular region.

Key words: PM$_{2.5}$, aerosol pH, NHx phase partitioning, meteorological parameters, E-AIM

## 1. Introduction

Aerosol acidity is among the most important parameters for atmospheric particulate chemistry as it has a large impact on both gas/particle partitioning of semi-volatile ionizable components and rates of many reactions occurring in aerosol liquid water (Seinfeld and Pandis, 2006;Losey et al., 2016;Guo et al., 2017a;Nah et al., 2018;Freedman et al., 2019). Laboratory and



field studies have also shown that enhanced acidity in aerosol can increase the formation of secondary organic aerosol (Surratt et al., 2007;Zhang et al., 2007b), one of the major components responsible for particulate air pollution (Zhang et al., 2007a;Huang et al., 2014). Studies have shown that highly acidic particles can have adverse health effects related to respiratory diseases (Utell et al., 1983;Dockery et al., 1996). More acidic particles can enhance the solubility of several trace metals in

fine particulate matter ($PM_{2.5}$) to induce oxidative potential inside the human body as an indirect health effect of acidic particles (Fang et al., 2017).

pH is the parameter frequently used to describe the acidity of aerosol liquid water and is calculated as the negative logarithm of hydrogen ion activity. Very few methods can directly measure the pH of aerosol (Rindelaub et al., 2016;Craig et al., 2018;Wei et al., 2018), so aerosol acidity is usually presented using indirect proxies such as ion balance (cation-to-anion

ratio), neutralization ratio ($[NH_4^+]/(2[SO_4^{2-}])+[NO_3^-]$) and strong acidity ($\Sigma cation-\Sigma anion$) (Yao et al., 2006;Du et al., 2010;Zhou et al., 2017). However, these indirect methods can lead to substantial uncertainty in acidity assessment (Hennigan et al., 2015;Guo et al., 2016;Murphy et al., 2017). Calculation of pH through thermodynamic modelling such as E-AIM (Clegg et al., 1998) and ISORROPIA II (Fountoukis and Nenes, 2007) with the input of reliable chemical compositions of the aerosol and at least one semi-volatile gases and meteorological data has been shown to be a more rigorous approach to calculate the

pH of aerosol liquid water. Murphy et al. (2017) and Song et al. (2018) both showed that the constraint from phase partitioning of $NH_3/NH_4^+$ should be included in calculations using aerosol thermodynamic models to get reliable pH calculations, which indicates that $NH_3$ observation can greatly improve the reliability of the aerosol acidity assessment.

However, long-term observations of $NH_3$ with corresponding particulate matter composition measurements are very scarce. Some of the few long-term monitoring networks are NAMN in the UK, SEARCH in the US and NAPS in Canada. In

the UK, 30 of the NAMN (National Ammonia Monitoring Network) sites have been providing monthly-integrated $NH_3$ and particulate inorganic ionic composition data since 2000 (Tang et al., 2018). SEARCH (Southeastern Aerosol Research and CHaracterization Network) is one long-term monitoring programs of $NH_3$ in the United States, which has both 24-hour average gas phase $NH_3$ and particle phase ionic species measurements at eight sites located in Southeastern United States with the sampling frequency of every three days. In Canada, the National Air Pollution Surveillance (NAPS) program is the only

program employing the long-term measurement of 24-hour average $NH_3$ concentrations with a sampling frequency of every three days at twelve monitoring sites where $PM_{2.5}$ constituents are also measured.

Several studies have examined the response of pH to changes in chemical composition with conceptual modelling. Weber et al. (2016) showed that in summertime, aerosol tends to remain highly acidic with the reduction of sulfate and the increase of ammonia-to-sulfate ratio. The modelling results of Guo et al. (2017b) showed that for wintertime meteorological conditions

aerosol pH has similar sensitivity to $NH_3$ concentrations in China and the eastern US under fixed temperature and relative

humidity. These two sensitivity tests of aerosol pH suggest that aerosol pH seems to have weak response to chemical composition changes. However, these studies both carried out conceptual modelling or calculations under fixed meteorological parameters. The time series of aerosol pH in six regions within the United States calculated by Lawal et al. (2018) showed that pH has seasonal variation that depends on the region, suggesting that changes in meteorological conditions can also contribute

to pH changes. Thus, the relative contributions of changes in meteorological conditions and chemical composition to variability in pH are still unclear and require systematic studies. As a result, in this study, we focus on the long-term variations of aerosol pH at six eastern Canadian sites (four urban and two rural) along with conceptual modellings of the sensitivity of aerosol pH to meteorological parameters and chemical composition changes to assess the major factors determining aerosol pH, and its variability, in different sites and seasons.

**2. Data sources and analysis method**

**2.1 NAPS database and chosen study sites**

All the $PM_{2.5}$ composition and trace gas data used in this study were downloaded from the NAPS program website (www.ec.gc.ca/rnspa-naps/). The NAPS program provides long-term monitoring of air quality-relevant pollutants at various sites across Canada. Altogether, there are twelve sites with comprehensive gas phase and particle phase composition

measurements, including four coastal sites (Halifax, Saint John, Burnaby and Abbotsford), one central site (Edmonton) and seven eastern sites (Toronto, Simcoe, Hamilton, Windsor, Ottawa, Montreal and St Anicet). 24-hour ambient samples of gaseous components (including $NH_3$ and $HNO_3$) and the dominant inorganic components of $PM_{2.5}$ were taken from 00:00-24:00 generally every three days.

Six sites in the provinces of Ontario and Quebec were chosen for this study (Figure 1), including Toronto (the largest city

in Canada), Ottawa, two other metropolitan cities (Windsor and Montreal) and two sites in rural regions (Simcoe and St Anicet). The details of these monitoring sites including their longitudes and latitudes, NAPS monitoring station codes, and duration of the data record used for this study can be found in Table S1. These sites were chosen because 1) they have long-term (at least 8 year) records of gaseous and $PM_{2.5}$ chemical components concentrations; 2) they are located in the central region of North America with little influence of sea salt (Vet et al., 2014) and therefore the acidity of $PM_{2.5}$ is mainly dominated by the

chemistry of sulfate, total nitrate and total ammonia (Young et al., 2013;Allen et al., 2015;Hennigan et al., 2015).

Measurements of 24 h-average concentrations between 2007 and 2016 of alkaline and acidic gaseous components and major water-soluble inorganic ions in $PM_{2.5}$ were used in this study for pH calculation. Citric acid-coated denuders were used to collect gas phase $NH_3$ samples. $Na_2CO_3$-coated denuders were used to collect gas phase $HNO_3$ samples. After alkaline and acidic gases were removed by passing through the two denuders, $PM_{2.5}$ samples were captured by a filter pack consisting of a



front Teflon filter and a back-up Nylon filter. The use of nylon filter is to capture volatized nitrate during sampling as the artefact of the evaporation ammonium nitrate and correct the data accordingly (Yu et al., 2006;Babich et al., 2011). All the extracts were measured by ion chromatography. Field blanks were routinely performed as the background corrections. The detailed protocol for ambient constituent measurements for NAPS database can be seen in Dabek-Zlotorzynska et al. (2011).

In total, there are 1067, 851, 711, 840, 713 and 742 sets of valid data used in the sites of Toronto, Ottawa, Windsor, Montreal, Simcoe and St Anicet, respectively.

Relevant historical meteorological data including hourly average temperature, relative humidity and atmospheric pressure were downloaded from http://climate.weather.gc.ca. The closest stations with complete data recordings were chosen as the meteorological data source. 24-hour average values of meteorological parameters during each sampling time period were
calculated and used as the input for E-AIM thermodynamic calculation. Specifically, the St Clair meteorological station was chosen as the meteorological data source for Toronto NAPS site (0.9 km from sampling site), the Delhi CS station for Simcoe site (21 km), the Windsor A station for Windsor site (9 km), the Ottawa CDA RCS station for Ottawa site (6 km), the McTavish station for Montreal site (16 km), and the St-Anicet 1 station for St Anicet site (0.1 km).

### 2.4 E-AIM thermodynamic modelling

The Extended AIM thermodynamic model (http://www.aim.env.uea.ac.uk/aim/aim.php) was applied for the calculation of aerosol acidity as E-AIM relies on few assumptions during the calculation of aerosol inorganic components behavior and it can also calculate the activity coefficient of each ion, which is crucial for aerosol pH calculation (Wexler, 2002). Specifically, model II, which mainly focuses on the thermodynamic calculations of $H^+$-$NH_4^+$-$SO_4^{2-}$-$NO_3^-$-$H_2O$ system (Clegg et al., 1998), was used in this study. Model II was chosen rather than model III or IV because it can deal with a wider range of relative
humidity and temperature, which gives more reliable information of the seasonal variation of aerosol acidity. The SNA (sulfate, nitrate and ammonium) components in $PM_{2.5}$ generally contributed more than 80% of total charges balance in the particles. Temperature, relative humidity, total ammonia (summation of gas phase ammonia and ammonium in $PM_{2.5}$), total nitrate (summation of gas phase nitrate acid and nitrate in $PM_{2.5}$) and sulfate were used as the inputs for this model. Partitioning of both $NH_3$/$NH_4^+$ and $HNO_3$/$NO_3^-$ allows for a strong constraint on the aerosol pH calculation (Murphy et al., 2017). Particles
were forced to be metastable without any formation of salts or ice precipitation. Aerosol pH was calculated according to the formula (Robinson and Stokes, 2002):

$$pH = -\log(a_{H^+}) = -\log(f_{H^+} \times \chi_{H^+} \times 55.509)$$

where $a_{H^+}$ is the activity of hydrogen ion in aerosol liquid water, $f_{H^+}$ is the hydrogen ion's mole-fraction based activity coefficient, $\chi_{H^+}$ is the mole-fraction of hydrogen ion, and 55.509 is the conversion factor from mole-based activity coefficient

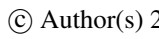



to molality-based one. Model IV was also used to assess the influence from non-volatile cations.

## 3. Results and discussion

### 3.1 Time series of pH in six Canadian sites

The 10-year time series of $PM_{2.5}$ pH values at the six sites is illustrated in Figure 2. To our knowledge, this is the first

long-term study of aerosol pH in Canada and provides one of the longest records for the evaluation of trends anywhere in the

world. The reliability of the pH calculations depends on several assumptions, including that daily average values are

appropriate for the calculations, and that the gas and particles phases are equilibrated. One rigorous method to evaluate the

reliability of the calculated pH is to compare the input (measured) and output (modelled) gas-particle partitioning of semi-

volatile species. The comparisons of modelled and measured gas phase fractions of total ammonia, denoted as

$Frac(NH_3)=n(NH_3)/[n(NH_3)+n(NH_4^+)]$, are plotted in Figure S2, where the linear regressions were all close to 1:1 line with

high $R^2$ (>0.88), indicating that the E-AIM modelling results are consistent with the observed phase partitioning characteristics

of $NH_3/NH_4^+$. The use of the phase partitioning of both $NH_3/NH_4^+$ and $HNO_3/NO_3^-$ to constrain the aerosol pH helps to avoid

significant over- or underpredictions (Murphy et al., 2017).

In Figure 2, all the time series of aerosol pH show a strong seasonal oscillation pattern between 1.5 to 4.0. By coloring

the pH values by ambient temperature, we can clearly see that particles were generally more acidic when ambient temperature

was higher. This variation pattern is consistent in both spatial (six sites) and temporal (10-year) terms. In the long-term, the

seasonal variation of aerosol pH fits well with the variation of ambient temperature. However, larger variation of aerosol pH

can be identified during wintertime. In some cases, the aerosol is very acidic, with pH<2, even when temperature <0 °C. This

variation pattern is more clearly illustrated in the box and whisker plots of pH in each month in six sites shown in Figure 3.

The pH values are lowest in the summer (July and August averages <2) and highest in the winter (January and February

averages > 2.5). The aerosol was consistently more acidic and had smaller pH variation in warmer months, and showed higher

average (and median) pH values but exhibited larger variation in wintertime.

To assess the potential influence of non-volatile cations including $Na^+$, $K^+$, $Mg^{2+}$ and $Ca^{2+}$ to the seasonal variation

pattern of aerosol pH, aerosol pH was also calculated using E-AIM model IV (Friese and Ebel, 2010) with the input of $Na^+$

concentration representing the contributions of all non-volatile cations ($[Na^+]+[K^+]+2[Mg^{2+}]+2[Ca^{2+}]$). The pH calculation

results are illustrated in Figure S3. Because E-AIM IV deals with narrower range of meteorological conditions (T $\geq$263.15 K,

RH $\geq$60%), E-AIM IV calculations were only possible for a limited part of the data sets. However, shown in Figure S3, E-

AIM IV modelling results show similar time series compared to E-AIM II results. These results confirm that the aerosol pH

seasonal variation pattern was not significantly affected by the non-volatile cations. The involvement of non-volatile cations



typically increases pH values by less than 0.1, and the change is less than 0.4 for more than 90% of data (shown in Figure S4).

The factors influencing aerosol pH can be roughly classified into two categories: meteorological parameters (including ambient temperature and relative humidity) and chemical composition (including gas phase and particle phase components). The following discussion will focus on the conceptual modelling of aerosol pH sensitivity to meteorological conditions in

comparison to the impacts of changes in chemical composition.

**3.2 Seasonal cycle of aerosol pH**

To assess the sensitivity of aerosol pH to meteorological conditions and chemical composition changes individually, we first examine the pH of aerosol with chemical composition held fixed at representative values under different meteorological conditions. Then we study the aerosol pH sensitivity to chemical composition changes under fixed typical meteorological

conditions of each of the four seasons. Figure 4 displays the aerosol pH calculated under every combination of relative humidity from 30% to 95% and temperature from -10 ºC to 30 ºC with fixed chemical composition. The chosen concentrations for sulfate, $TNO_3$(=$HNO_3$+$pNO_3$) and $NH_x$ are 15 nmol m$^{-3}$, 30 nmol m$^{-3}$ and 180 nmol m$^{-3}$ $NH_x$, respectively, which are close to the ten-year average concentrations of chemical composition in Toronto atmosphere (which were 17.6 nmol m$^{-3}$ sulfate, 34.2 nmol m$^{-3}$ $TNO_3$ and 188 nmol m$^{-3}$ $NH_x$).

It can be seen in Figure 4 that particles tend to become more acidic at higher temperature and lower relative humidity and become more neutralized when temperature is lower or RH is higher. The square symbols on Figure 4 are the monthly average values of ambient temperature and relative humidity in Toronto from 2007-2016 with standard deviations as error bars. The seasonal cycles of meteorological conditions in five other sites are listed in Table S2, and generally show similar seasonal cycling of RH and temperature. Even excluding changes in chemical composition, we find from Figure 4 that we can explain

the summer minimum and winter maximum in pH from Figure 3. Temperature is the main factor driving the seasonal variation of aerosol pH while the changes due to RH variation are much smaller. This result suggests the central role of meteorological conditions, especially temperature, in the determination of aerosol pH seasonal cycle in mid- and high-latitude regions.

The temperature dependence of aerosol pH can be theoretically derived from the phase partitioning $NH_3$/$NH_4^+$ based on the equilibrium: $NH_3$(g) ↔ $NH_3$(aq) and $NH_4^+$ ↔ $NH_3$(aq) + H$^+$, which is governed by $K_H$ and pKa, respectively (Hennigan

et al., 2015). Because both $K_H$ and pKa have strong temperature dependencies (Chameides, 1984;Bell et al., 2007), aerosol pH is going to temperature dependent even if liquid water content or $NH_3$/$NH_4^+$ partitioning behavior does not change. The partial derivative of aerosol pH dependence on temperature will give $\partial pH/\partial T \approx -0.05$ (K$^{-1}$), which corresponds to 0.1 unit increase (decrease) of aerosol pH if temperature decreases (increases) by 2 ºC.

It is possible that the sensitivity of aerosol pH to chemical composition is different in different seasons. As a result, the

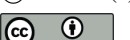



sensitivity of aerosol pH to composition changes in each season is further studied through the conceptual modelling. In Figure 5(a)-(d) the pH calculated under different combinations of $NH_x$ and sulfate concentrations under typical temperature and RH conditions in spring (April, T=10 °C, RH=60 %), summer (July and August, T=25 °C, RH=70 %), fall (October, T=10 °C, RH=75 %) and winter (January and February, T=-5 °C, RH=75 %) in the Toronto atmosphere are shown. Total nitrate

concentrations were set to be two times sulfate concentrations. This assumption is representative of the Toronto atmosphere as ten-year average of sulfate concentration (17.6 nmol m$^{-3}$) is approximately half of total nitrate molar concentration (34.2 nmol m$^{-3}$) though for any given day the ratio could be different. The corresponding calculation results of particle phase fraction of nitrate $\varepsilon(NO_3^-)$ in four seasons is shown in Figure S5. During summer ammonium nitrate formation is unfavorable, while in wintertime ammonium nitrate can form very efficiently. The effect of nitrate formation to aerosol pH will be further discussed

in section 3.4.

The markers on Figure 5 represent the median and the range (from 10th percentile to 90th percentile) of $NH_x$ and sulfate concentrations in four seasons in six monitoring sites from 2007 to 2016. The three more westerly sites (Toronto, Windsor and Simcoe) tend to have higher sulfate in all seasons and Toronto has slightly higher $NH_x$. However overall all six sites occupy regions of similar chemical sensitivity in the plots. The conceptual modelling of aerosol pH under summertime condition

shown in Figure 5(b) suggests that the relatively low variability in summertime pH shown in Figure 3 results from the aerosol pH being insensitive to chemical composition changes under the corresponding meteorological conditions. Specifically, in summertime, pH was very constrained to the small range around 2 with little influence from the variation of chemical composition, even though the sulfate loadings were the most variable in the summer. Judging from the concentration ranges of sulfate and $NH_x$ shown in all six sites, in summertime even decreasing sulfate concentration by one order of magnitude do

not significantly change the pH values, and this effect became more obvious with higher $NH_x$:sulfate molar ratio.

By comparison, the pH calculated under spring and fall temperature (10 ºC) shown in Figure 5(a) and 5(c) showed that pH became more sensitive to $NH_x$ variation but still remain quite insensitive to sulfate concentration changes within the relevant concentration range. In spring, the median concentrations of $NH_x$ in the sites of Toronto, Simcoe and Windsor were higher compared to the other three sites, leading to average April pH values in Toronto, Simcoe and Windsor (2.40~2.69) that

were consistently higher than the other three sites (2.15~2.31). In October, $NH_x$ and sulfate concentrations had wider ranges of concentration, but we can conclude from Figure 5(c) that the pH changes due to chemical composition were mainly driven by the variation of $NH_x$. The effect of relative humidity changes can be seen from the comparison between Figure 5(a) and 5(c), where it shows that with fixed chemical composition, higher RH will slightly make aerosols less acidic.

By comparison, under wintertime meteorological conditions, aerosol was more neutralized in the region $NH_x$:sulfate>2.

Apart from the effect of lower temperature, lower concentration of $NH_x$ and lower $NH_x$ to sulfate molar ratio also made aerosol

pH much more sensitive to chemical composition changes than the other seasons. This could contribute to the larger variation of wintertime aerosol pH shown in Figures 2 and 3. In Simcoe, Windsor and St Anicet, the sites with lower $NH_x$:sulfate molar ratios, wintertime pH values all exhibited significant variations. Under these conditions, not only the uncertainty in the SNA concentration measurements, but also the potential contributions from organic matter such as organic ammonium salts (Schlag

et al., 2017;Tao and Murphy, 2018) or organosulfates (Vogel et al., 2016;Glasius et al., 2018), which current thermodynamic models do not fully consider, can contribute to uncertainty in aerosol pH calculation. These conceptual modelling results also provide insight into the challenges posed in the direct measurement of aerosol pH off-line as it is both strongly impacted by ambient environmental conditions and the $NH_3$ concentration equilibrated with the particle phase.

**3.3 Long-term trends of pH in summertime**

10       The above time series analysis and conceptual modellings of aerosol pH suggest that in the study region during summertime, aerosol pH is strongly impacted by ambient temperature with much weaker response to the other factors while during wintertime aerosol pH is both affected by temperature and chemical composition. Figure 6 shows the summertime (data from July and August) median aerosol pH in each year in the six monitoring sites. Summertime average temperature was also plotted on right axes in each graph. The clear anti-correlation indicates that interannual variability in summer temperatures

causes interannual variability in aerosol pH. Another application of this result can be found in Battaglia et al. (2017), who showed that urban aerosol in U.S. cities tended to be more acidic than rural aerosol due to urban heat island effect, consistent with the important role of temperature to aerosol pH determination.

        The study conducted by Weber et al. (2016) found that in summertime, aerosol remained highly acidic even with significant reduction of sulfate in the US. A similar conclusion could be drawn for conditions in Canada, where we found the

role of temperature to be more significant than variations in chemical composition for trends in aerosol pH. Significant changes in the particle phase sulfate have been observed during the summers of 2007-2016. Figure 7 displays the summertime average (data from July and August) concentrations of $PM_{2.5}$ sulfate at the six sites. Linear regressions are used to show the annual reductions of fine particulate sulfate. Significant decreasing trends (p-value of linear regressions <0.05) of sulfate have been observed in five out of six sites (excluding the St Anicet site where large inter-annual variation was found). Specifically, sulfate

concentrations have experienced 61%, 56%, 62% and 67% reductions in Toronto, Simcoe, Windsor and Ottawa, respectively, over the decade from 2007 to 2016. There is no statistically significant long-term trend in the NHx values at any of these sites. Summertime aerosol pH is not decreasing with the decreasing trend of sulfate but is mainly influenced by the inter-annual variation of meteorological conditions.

**3.4 pH trends in wintertime**


To assess the effect of chemical composition changes on aerosol pH in winter meteorological conditions, wintertime average concentrations of sulfate and $NH_x$ in each year in each site were plotted on Figure 8(a) with the color representing the corresponding average pH values. The background color in Figure 8(a) represents pH calculated under different combinations of sulfate and $NH_x$ concentrations with fixed temperature (-5 °C), RH (75%) and sulfate to $TNO_3$ molar ratio (1:2), the same

as Figure 5(d) but with a different scale and focused on a smaller range of chemical conditions. The figure clearly shows that changes in chemical composition under wintertime meteorological conditions can lead to significant changes in aerosol pH. Changes of 20 nmol $m^{-3}$ in either $NH_x$ or sulfate can change aerosol pH by more than 0.5 units for some cases. Also, although most of the wintertime average pH values calculated for the actual chemical composition at each site (colors of the markers) generally agree with the pH calculated as the conceptual model (colors of the same position in the background), some of the

data points were slightly more acidic than the background value, and this is likely the result of our simplified assumption of fixed sulfate to $TNO_3$ molar ratio during the calculation in the conceptual modelling.

Figure 8(b) shows the effects that temperature, relative humidity and $TNO_3$ concentrations can have on the pH of aerosol with 10 nmol $m^{-3}$ sulfate under wintertime meteorological conditions. Generally, lower temperature and higher relative humidity will make aerosol more neutralized. However, the effect of the addition of total nitrate is more complicated. When

the added $TNO_3$ concentration is small, aerosol pH shows an increasing trend with $TNO_3$ concentration. With the continuous increase in the $TNO_3$ concentration, aerosol will become more acidic after a turning point. Noticeably, when aerosol pH shows a decreasing trend with $TNO_3$ concentration, there is still excess $NH_3$ in the gas phase. Figure S6 shows the box and whisker plots of wintertime average molar ratio of $TNO_3$ to sulfate concentration in each year in each sampling site, where it shows that the molar concentration of $TNO_3$ was generally higher than 2 times of sulfate concentration, which could contribute to

wintertime aerosol being more acidic than the value calculated forcing sulfate:$TNO_3$=1:2. As a result, during wintertime, both chemical composition and meteorological conditions can have significant impacts on aerosol pH without a clear dominant factor. As a corollary, the uncertainty in measurements of particle composition, the influence of non-volatile cations and the potential contribution from organic acids or organosulfates will also have a more significant impact on the determination of aerosol pH in winter by potentially over- or underestimating the sulfate or nitrate that is balancing the ammonium.

**4. Conclusion**

The pH values of fine particulate matter at six eastern Canadian sites, including Toronto, Windsor, Simcoe, Ottawa, Montreal, and St Anicet from 2007 to 2016 were calculated by E-AIM model constrained by input from the NAPS database and meteorological data. Strong seasonal cycles of aerosol pH were found in all six sites over the ten years. A consistent pattern of aerosol pH fluctuation from acidic (pH <2) in summer to higher pH around 3 in winter was found at all six sites. Ambient

temperature proved to be mainly responsible for the seasonal cycle of aerosol pH. The sensitivity tests of pH to chemical



composition shows aerosol pH has different sensitivity to chemical composition changes in different seasons. During summertime, aerosol pH was mainly controlled by ambient temperature while the fluctuation of $NH_x$ and sulfate concentrations did not contribute significantly to the inter-annual variation of aerosol pH. In comparison, in wintertime, aerosol pH can be greatly influenced by both chemical compositions and meteorological conditions. When temperatures are low, aerosol pH at

these six sites are very sensitive to all the changes in $NH_x$, sulfate and $TNO_3$ concentrations, leading to the higher variation of aerosol pH in wintertime compared to that in summertime. This study focused on a number of sites with relatively low ambient mass loadings of aerosol inorganic constituents. However, for places with high particulate pollution, similar approaches can also be applied to assess the aerosol pH sensitivity to chemical composition and meteorological parameters to assess the generalizability of our findings.

**Data availability**

All the data and model used in this study are publicly available, including chemical composition data (http://maps-cartes.ec.gc.ca/rnspa-naps/data.aspx), meteorological data (http://climate.weather.gc.ca/) and E-AIM modelling (http://www.aim.env.uea.ac.uk/aim/aim.php). Further relevant information can be obtained upon request on corresponding author.

**Author contributions**

YT performed all the data processing and modelling calculation. YT and JGM analyzed data, drew the conclusions and wrote the manuscript together.

**Acknowledgement**

The authors thanks Enyu Xiong for the help of organizing data. Ye Tao is supported by Ontario Connaught Scholarship

from the University of Toronto.

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



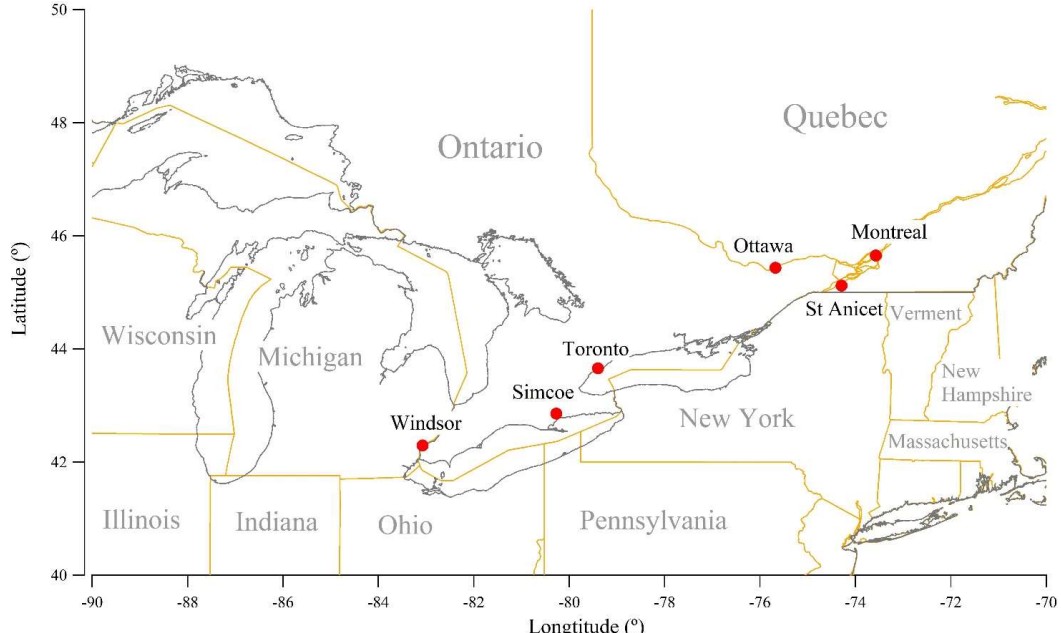

**Figure 1. Six NAPS monitoring sites chosen for this study: Toronto, Ottawa, Windsor, Montreal, Simcoe and St Anicet.**





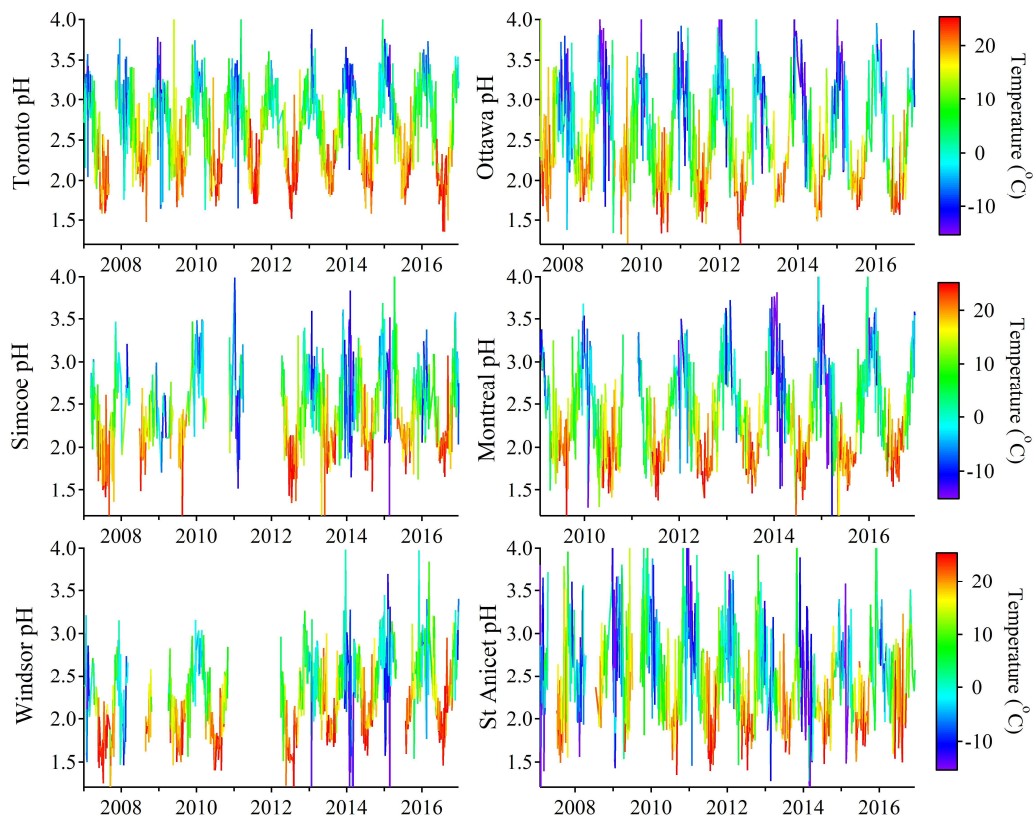

**Figure 2. The time series of PM$_{2.5}$ pH at six sites colored by ambient temperature.**





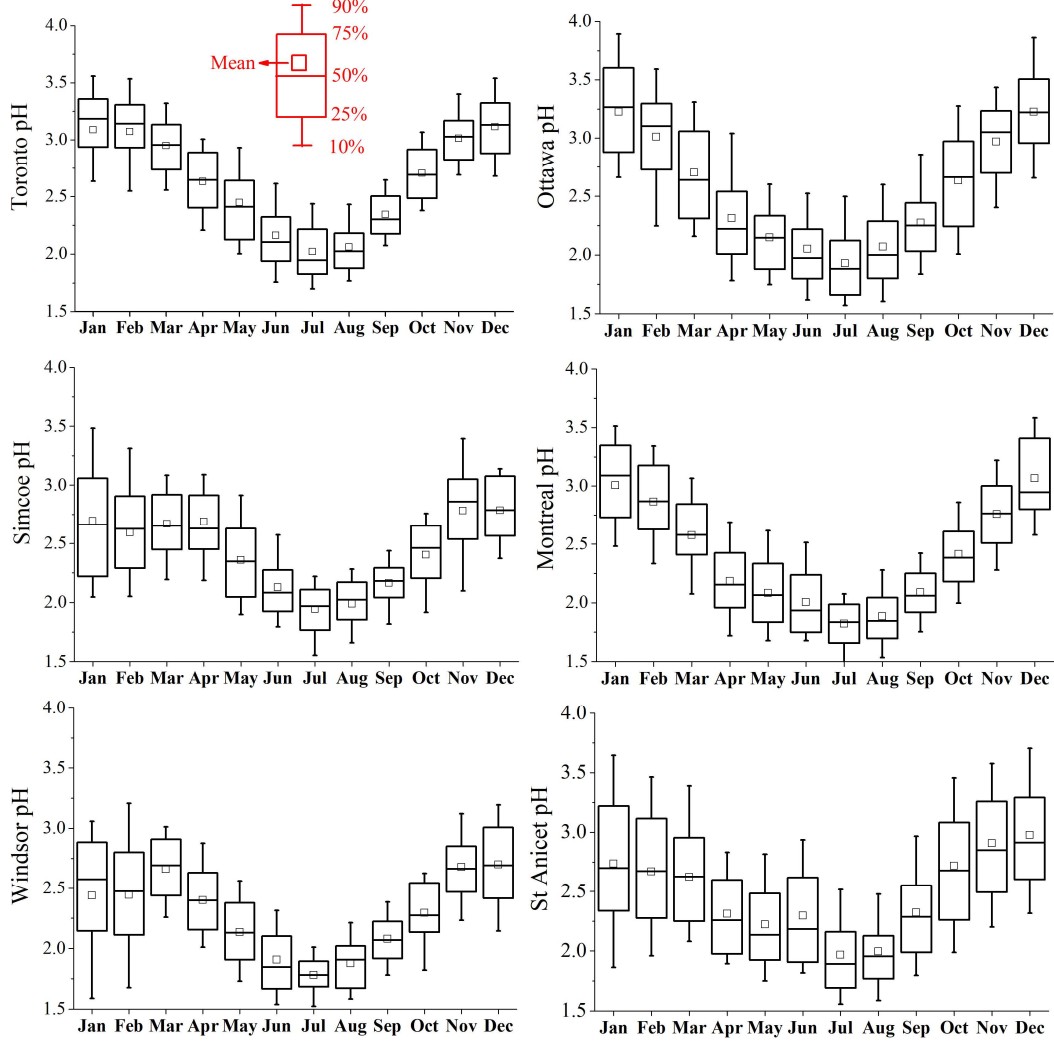

**Figure 3. The box and whisker plot of aerosol pH in each month in six sites. In each box, the top, middle and bottom lines represent 75th, 50th and 25th percentile of statistical data, the upper and lower whiskers represent 90th and 10th percentile, and the square mark represent the mean value.**



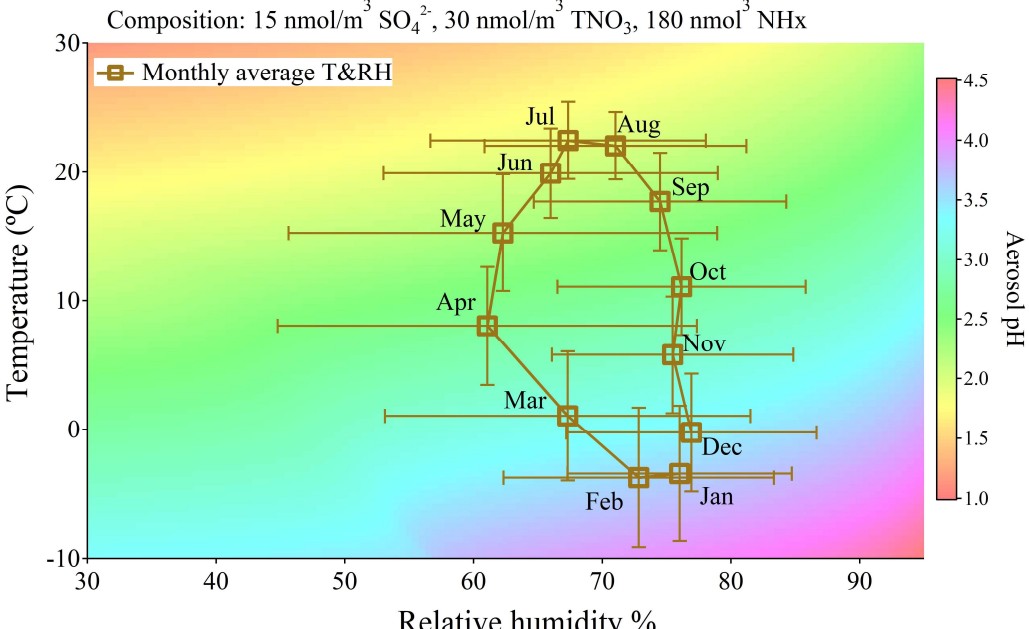

**Figure 4. Plot of E-AIM calculated pH under different combinations of relative humidity (from 30%-95%) and temperature (from -10 °C to 30 °C) with the fixed input chemical compositions (15 nmol m⁻³ sulfate, 30 nmol m⁻³ TNO₃ and 180 nmol m⁻³ NHₓ). The square plots on the graph represent monthly average of meteorological parameters in Toronto from 2007-2016 with standard deviations as error bars.**





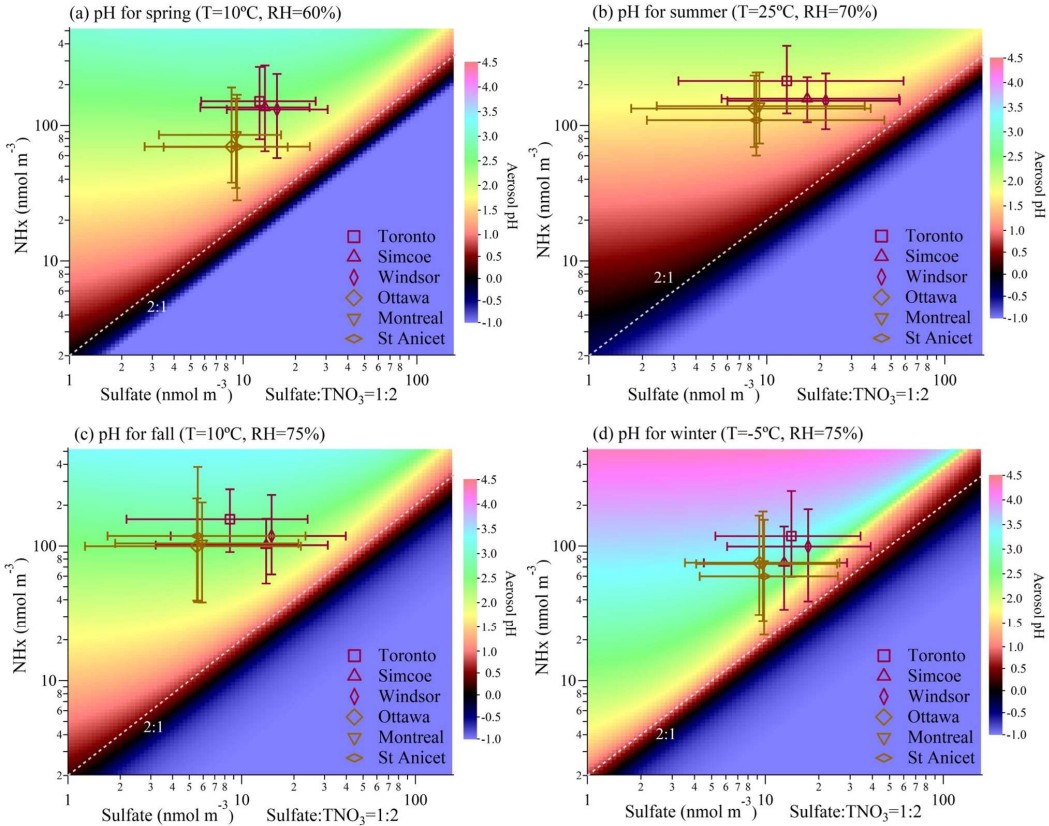

**Figure 5. Aerosol pH calculated with fixed meteorological parameters (temperature and RH) under different combinations of NHₓ and sulfate concentrations. Total nitrate concentrations are set to be 2 times that of sulfate. The markers on the graph represent median summertime concentrations of NHₓ and sulfate in six sites from year 2007 to 2016 with error bars indicating 10th percentile to 90th percentile. The fixed meteorological parameters chosen in Figure (a)-(d) are representative of the conditions in spring (April), summer (July and August), fall (October) and winter (January and February) shown in Figure 4.**





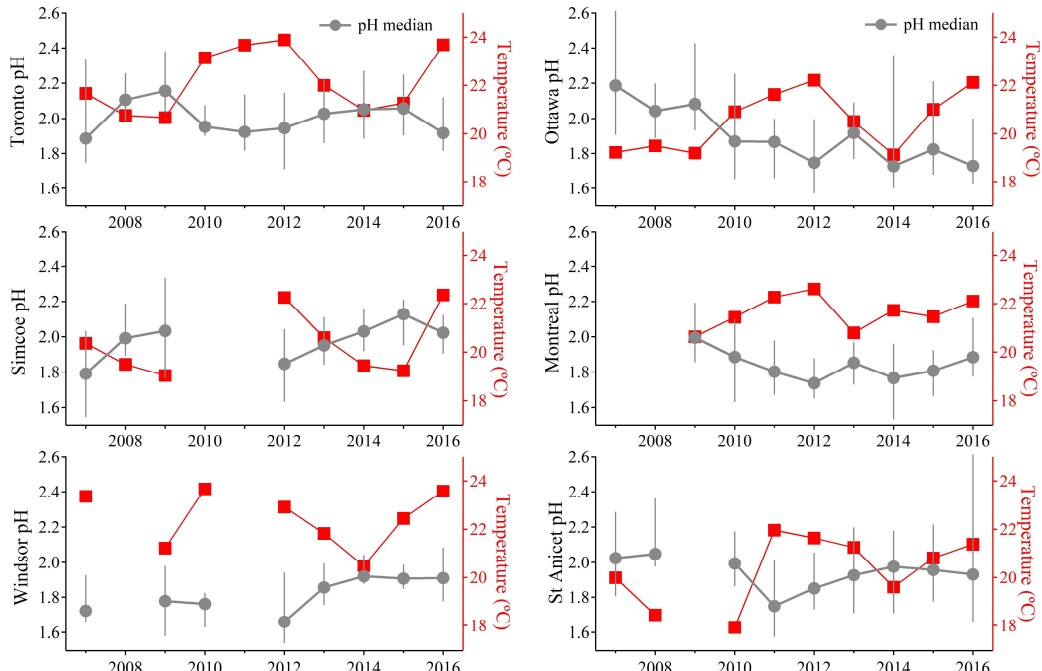

**Figure 6. Median of aerosol pH in summertime (data on July and August) from year 2007 to 2016 in six monitoring sites. The error bars with median pH values represent the range from 25% to 75% percentile. Average ambient temperature in summertime (July and August) in each site is also plotted on right axis to illustrate the reverse trend of temperature with pH inter-annual variation.**





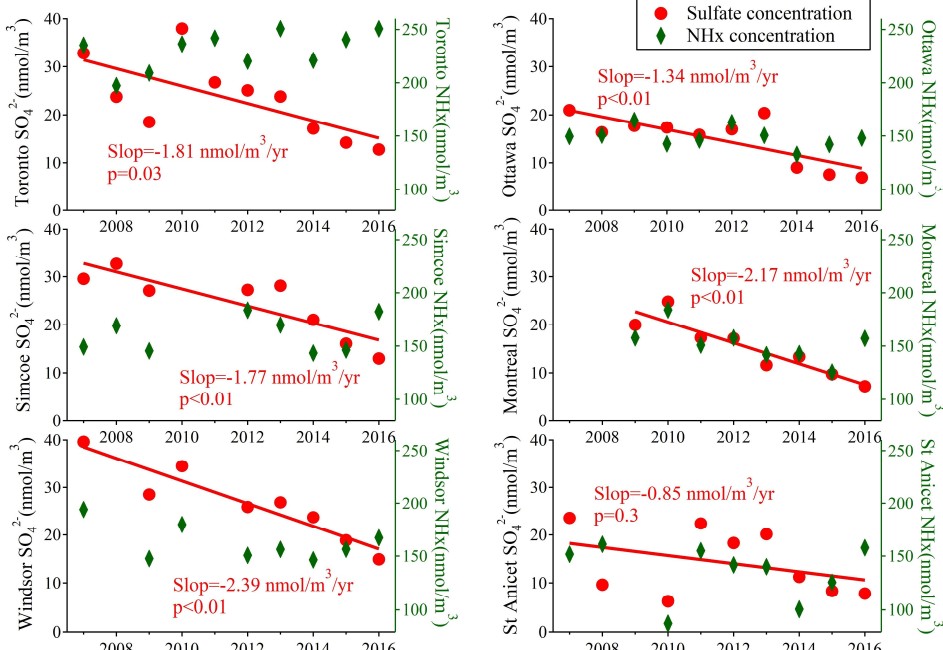

**Figure 7. 10-year trends of summertime average concentrations of sulfate and NH_x in each site. Linear regressions of sulfate were illustrated on each graph to represent the reduction of particulate sulfate from 2007 to 2016.**



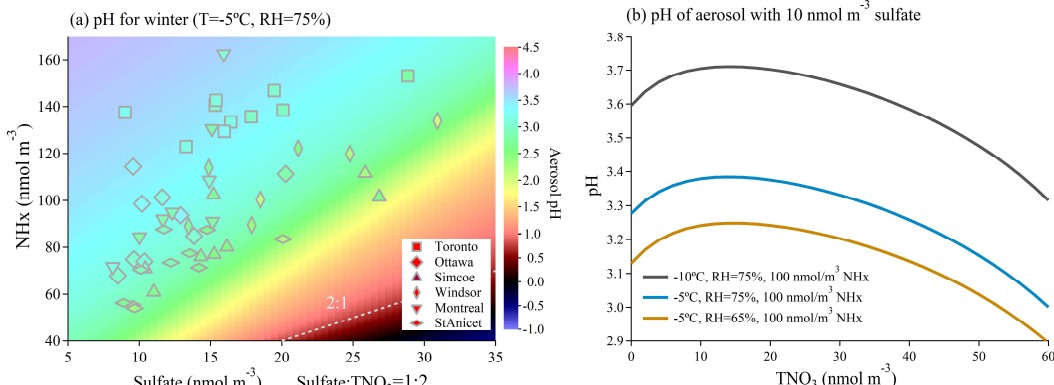

**Figure 8. (a) The plot of aerosol pH under wintertime meteorological conditions (-5 ºC, RH=75%) under different combinations of sulfate and NH$_x$ with more focused region compared to Figure 5(d). The markers on the graph represents the average wintertime (data from January and February) sulfate and NH$_x$ concentrations in each year in each sampling site with color indicating average pH value. (b) pH of aerosol with 10 nmol/m$^3$ sulfate under different combinations of temperature and relative humidity as the function of TNO$_3$ in the system.**