# Peer review of "The sensitivity of PM2.5 acidity to meteorological parameters and chemical composition changes: 10-year records from six Canadian monitoring sites"

_Atmospheric Chemistry and Physics, 2019_

## Referee Comment (RC1) · Anonymous Referee #1 · 3 Apr 2019

Review of "The sensitivity of PM$_{2.5}$ acidity to meteorological parameters and chemical composition changes: 10-year records from six Canadian monitoring sites" by Tao and Murphy

**General Comments:**

This paper presents aerosol pH predictions at six Canadian sites based on long-term aerosol composition and NH$_3$ monitoring data. The authors characterize temporal trends in aerosol pH and seasonal averages at each site. Using the observational data, the authors characterize the sensitivity of pH to meteorology (Temp and RH) and composition (sulfate-nitrate-ammonium). The authors find that pH has a distinct seasonal profile at all of the sites, with summertime pH systematically lower than wintertime pH. During summer, aerosol pH is much less sensitive to perturbations in composition than it is to the met parameters. During winter, aerosol pH is still quite sensitive to meteorology, but is more sensitive to compositional changes than during summer. Overall, these results are novel and likely to be of interest to many in the atmospheric chemistry community. The topic is certainly relevant for *ACP*. The writing is clear, and the conclusions are robustly supported by the measurements and modeling results. I strongly endorse the manuscript for publication after the following specific concerns are addressed.

**Specific Comments:**

It is important that the authors state that their findings only apply to the sulfate-nitrate-ammonium system. Locations with dust and/or marine influence may exhibit very different trends and sensitivities to meteorology and composition.

The authors should discuss possible sampling artifacts as a function of temperature and season. The discussion on Pg. 4, lines 1-2 is presented in a way that assumes no sampling artifact, but studies (including the Yu et al. (2006) study that the authors cited) show evaporation of NH$_4$NO$_3$ from nylon filters. There is a temperature and RH-dependence to these losses, which are often greatest in summer. How would even small sampling artifacts during summer influence these results?

How were solids treated (prohibited or allowed?) in the model runs? How were data points with very low RH and ALW treated? Presumably, there were more of these points during winter, when dry, artic air influences many of the sites? The interpretation of pH under conditions with a low fraction of ALW seems questionable.

The authors should replicate Figure S2 with the predicted and measured $\varepsilon(NO_3)$.

While the authors find only minor effects of non-volatile cations (NVC) in their analysis, the limitations of this analysis need to be discussed. For example, Ca and Mg salts are typically much less soluble than analogous Na salts – replacing Ca and Mg with equivalent Na therefore does not account for potential precipitation, which may have a more pronounced effect on pH. Na is also more hygroscopic, which will affect the predicted ALW. Also, a brief discussion on the effects of NVC on pH (see Fig. 4c of Guo et al., 2018) should be included in the intro where composition effects on pH are discussed.

Temperature and RH are strongly linked in the atmosphere: discussing them as if they are independent parameters is not correct. A missing element from this manuscript is the role and trends in ALW, which is also closely connected to T and RH. Nguyen et al. (2016) show strong seasonal trends in ALW in the Northeast USA (see Fig. 2d), which would presumably have similar behavior to the Canadian sites analyzed in the present manuscript. If the Canadian sites do indeed show similar behavior (ALW maxima in summer), then it would imply the more acidic summertime particles occur despite the diluting effect of the ALW enhancement. This would be a very interesting contrast to the results of Guo et al. (2015) and Battaglia et al. (2017), who both show that aerosol pH decreases with increasing T. In their studies, the increase in T (and decrease in RH) that happen in the afternoon lead to decreasing ALW and a concentrating effect on $H^+$ (hence, decreasing pH). Clearly, the seasonal changes in composition would be an important factor that also contributes to these observations.

I found the explanation of Fig. 8b (pg. 9, line 12-24) to be quite confusing.

I completely disagree with the first sentence in the Abstract. Even if the chemical composition of an aerosol is known precisely, it is not possible to calculate the pH from an ion balance due to the buffering effects the $HSO_4^-/SO_4^{2-}$ equilibrium (similar effect for organic acids, as well). Further in the manuscript – pg. 2 line 12 – the authors cite several studies which also contradict this statement. To my knowledge, there has never been a study which demonstrated a good connection between the ion balance and pH, so I think it is a mistake to lead the manuscript with this idea.

**Technical Corrections:**

Pg. 1, line 10 "constitute" or "constituent"?

Pg. 2, line 7: say "aerosol aqueous phase" instead of "aerosol liquid water"

Pg. 2, line 8: suggest deleting "very"

Pg. 6, line 26-28: this seems important - explain the derivation of this theoretical relation. Also, can the authors compare the prediction to their data?

Pg. 7, line 29 (pg. 9, line 14): I suggest alternate wording ("aerosol was more neutralized") – because the pH is still quite acidic, suggest using "aerosol pH was higher" or similar.

Pg. 10, line 6-7; why would aerosol loading affect the results?

The role of organosulfates (mentioned pg. 8, line 5 and pg. 9, line 23) in aerosol pH is not clear? Can the authors add clarification?

The color scale on Figures 4, 5, and 8 can be difficult to distinguish certain levels – the color scale used with Fig. 2 is much easier to discern.

Address grammar and/or awkward phrasing in the following places: pg. 6, line 26; pg. 7 line 28; pg. 7 line 30.

**References:**

Battaglia, M. A., Douglas, S., and Hennigan, C. J.: Effect of the Urban Heat Island on Aerosol pH, Environmental Science & Technology, 51, 13095-13103, 10.1021/acs.est.7b02786, 2017.

Guo, H., Xu, L., Bougiatioti, A., Cerully, K. M., Capps, S. L., Hite, J. R., Carlton, A. G., Lee, S. H., Bergin, M. H., Ng, N. L., Nenes, A., and Weber, R. J.: Fine-particle water and pH in the southeastern United States, Atmos Chem Phys, 15, 5211-5228, 2015.

Guo, H., Nenes, A., and Weber, R. J.: The underappreciated role of nonvolatile cations in aerosol ammonium-sulfate molar ratios, Atmos. Chem. Phys., 18, 17307-17323, 2018.

Nguyen, T. K. V., V. P. Ghate, and A. G. Carlton (2016), Reconciling satellite aerosol optical thickness and surface fine particle mass through aerosol liquid water, Geophys. Res. Lett., 43, doi:10.1002/2016GL070994.

Yu, X. Y., Lee, T., Ayres, B., Kreidenweis, S. M., Malm, W., and Collett, J. L.: Loss of fine particle ammonium from denuded nylon filters, Atmospheric Environment, 40, 4797-4807, 10.1016/j.atmosenv.2006.03.061, 2006.

---

## Referee Comment (RC2) · Rodney Weber (Referee) · 15 Apr 2019

This paper determines particle pH over a 10-year period at a number of sites in eastern Canada. pH is reported to be higher in winter vs. summer by about one pH unit, similar to what has been reported in Atlanta, although the Atlanta aerosol pH is a bit lower in both seasons [Guo et al., 2015]. The authors conclude that the difference in pH is largely due to T. Another major finding is that pH is reported to be more variable in winter due to particle composition effects. The topic is appropriate for publication in ACP. The paper is an important contribution to the growing knowledge on fine particle

pH.

There are a number of points the authors could consider.

1) Although it is concluded that T is the main driver of lower pH in summer, with minor influence by aerosol composition, ambient data did not seem to be used in the assessment. For example, maybe a comparison table could be made showing summer vs winter average concentrations of the main species involved (sulfate, ammonium and nitrate). (I would also suggest including the ratio of nitrate to sulfate, which may be important for the next point.) Note that Guo et al (2015) reported a role of higher ion concentrations (sulfate) in summer as a cause for their reported differences in summer and winter pHs, but did not do a detailed investigation specifically on the effect of temperature. This paper can address this issue in much more detail and provide a more definitive conclusion. As noted in this paper, some of the T effects result from the equilibrium constants (Henry's law and dissociation, Ka's) depending on T. Guo et al (e.g., see [Guo et al., 2017] supplemental material: https://www.atmos-chem-phys.net/17/5703/2017/acp-17-5703-2017-supplement.pdf) has shown one way to look at T (and particle water effects) is through S curves of partitioning vs pH. I think this is a useful way to think of things and it may be useful here. Overall, I think more insight could be gained if the authors tried to remove or account for the T effects in their data and then look for other possible drivers of pH differences.

2) The contrast in particle chemical composition effect on pH could be explored further by investigating the role of particle nitrate, or with nitrate to sulfate ratios. The paper tends to hint at the role of nitrate on this effect, but maybe it is playing a very large role here. If so, the paper could be more precise than just stating composition effects. For example, in summer the nitrate levels are likely to be low because of the higher temperatures favoring the gas phase, resulting in low NO3-/SO42- ratios. Particle water, in the summer, is then largely controlled by the main hygroscopic species, non-volatile sulfate (and T, RH). pH depends on the inverse of particle water concentration. In contrast, in winter, the NO3-/SO42- ratio is likely much higher since nitrate partitioning to

the particle is more favored. (I.e., at lower T, there is a higher predicted fraction of NO3- to TNO3 at a given pH; see LWC and T affect on nitrate partitioning vs pH S curves in Guo et al. (2017) supplemental material). Indeed, Fig S5 shows that based on this data set, a small fraction of nitrate partitions to the aerosol in summer, whereas a much larger fraction is in the particle phase in winter. This means that in winter liquid water is affected by both sulfate and nitrate, and possibly more so by nitrate. Nitrate is semi-volatile, which leads to feed backs [Guo et al., 2017]. The last figure shows part of this dependence; initially the uptake of nitrate leads to higher liquid water (nitrate is highly hygroscopic), which dilutes the H+ and raises pH. If nitrate partitioning is in the sensitive part of the S curve then higher pH leads to more nitrate, which allows for more uptake of nitrate (see again S curves for nitrate partitioning and LWC effects). Hence the more partitioning of nitrate drives the pH higher (as seen in Fig 8b, although this is overwhelmed at some point apparently by the increase in H+ from the dissociation of HNO3 aq). Also, with high particle nitrate the system becomes more sensitive to T, and so overall it might simply be the higher nitrate levels in winter cause greater variability in winter pH.

Minor Comments.

First line of Abstract that states: Aerosol pH is difficult to measure directly but can be calculated if the chemical composition is known with sufficient accuracy and precision to calculate the aerosol water content and the H+ concentration through ion balance. Strictly speaking this is true, but in practical terms can it be done. For example, there is the issue of measurement uncertainty and propagation of those errors, leading to high uncertainty in H+. One must also measure all ions at actual ambient aerosol liquid water concentrations, eg, one would have to measure bisulfate and sulfate levels that exist in the ambient aerosol accurately. Given that ion balances have been used in the past to incorrectly infer acidity, I suggest removing this sentence to avoid perpetuating this idea? Second line of Abstract that states: In practical terms, simultaneous measurements of at least one semi-volatile constitute, e.g. NH3 or HNO3, are required to

provide a constraint on the calculation of pH. Is this strictly true as a general rule? Consider the sigmoid curves for the fractional partitioning of a semi-volatile species vs pH (see Guo et al. 2017 supplemental material). For a given set of conditions, there is only about a 1.5 to 2 pH range that partitioning is sensitive to pH, outside this range there is very little sensitivity (e.g., either all is in gas or all in particle phases). This line in the text could be clarified by stating that the semi-volatile constitute used to constrain the pH calculations should be in both the particle and gas phase, e.g., epsilon (=p/(g+p) or g/(g+p) ) between say 20 and 80%. It then raises the question of how one actually goes about solving for a final pH given missing species (i.e., does one iterate, is for how many times...). Line 15, as noted above, measured NH3/NH4+ constraint on pH only works when there is some fraction in each phase. If the the particle pH is very low or high, then it is possible for the system of have practically no gas phase NH3 present or no particle phase NH4+ present, in which case the observed concentrations of these species do not provide a good constraint on pH. Section 2.4: Some things to consider briefly discussing regarding the model and its assumptions: Lack of consideration of other species by E-AIM (e.g., non-volatile cations, which is discussed later in the paper), specific RH range data in this study (can one believe the predictions down to 30% and greater than 90% RH – why not test with comparison of observed/measured NH3 partitioning?), no consideration of organic species influence on LWC, and no consideration of possible phase separations (ie, more important at lower RH). Some of these issues are noted at the very end of the paper, it might be worth point them out when the thermodynamic model is discussed.

This paper often refers to an earlier paper by Murphy et al. 2017 Faraday Disc. In that paper the pH was calculated differently; there is a difference of a factor of 55.509, the conversion from mole-based to molality-based activity. Does this account for the generally higher pH reported in Murphy et al of 2.5 to 5.5 for a similar region as discussed in this paper (ie, that paper reported molarity based pH, whereas this paper molality based pH)? Also see [Jia et al.]. I think the difference in pH calculations (and results) between these two papers should be noted.

Page 5 and Fig S2. The comparisons between measured and predicted NH3 and NH4+ are very useful, but why not show a similar set of plots in the supplementary for HNO3 and NO3-. This would provide a more complete assessment of the model predictions and a useful contrast to what has been reported in a number of other papers which show that ammonia/ammonium partitioning is much better predicted by thermodynamic models than nitrate partitioning [Guo et al., 2017; Guo et al., 2016], [Guo et al., 2018] supplemental material, and [Nah et al., 2018] supplemental material.

Bottom of page 5 and on. It should be discussed that there is an important limitation in assessing the effect of non-volatile cations on pH using an equivalent Na+. Some non-volatile cations, such as Ca2+ and Mg2+, have greater nonlinear effects on activity than Na+ such that they can behave very differently than Na+ (e.g., CaSO4 can precipitate out). Because this is not considered, the effect of these cations may be greater than anticipated from the results based on an analysis using equivalent Na+.

Pg 6 Line 26 typo.

Pg 7 line 20 As noted above, one may want to point out this is due to LWC is controlled by non-volatile sulfate in the summer (which is likely not true in winter) and NH4+ volatility acts to buffer the pH of this system when there is little nitrate (Weber et al, 2017).

Pg 7 line 30 to Pg 8 line 2. Can a physical explanation be given to support the statement that lower NHx and lower NHx to sulfate molar ratio is the reason? Maybe it is nitrate that is playing a role here also?

Guo, H., R. Otjes, P. Schlag, A. Kiendler-Scharr, A. Nenes, and R. J. Weber (2018), Effectiveness of ammonia reduction on control of fine particle nitrate, Atm. Chem. Phys., 18, 12241-12256. Guo, H., J. Liu, K. D. Froyd, J. Roberts, P. R. Veres, P. L. Hayes, J. L. Jimenez, A. Nenes, and R. J. Weber (2017), Fine particle pH and gas-particle phase partitioning of inorganics in Pasadena, California, during the 2010 CalNex campaign, Atm. Chem. Phys., 17, 5703-5719. Guo, H., A. P. Sullivan, P. Campuzano-Jost, J. C.

Schroder, F. D. Lopez-Hilfiker, J. E. Dibb, J. L. Jimenez, J. A. Thornton, S. S. Brown, A. Nenes, and R. J. Weber (2016), Fine particle pH and the partitioning of nitric acid during winter in the northeastern United States, J. Geophys. Res. Atmos., 121(17), 10,355-310,376. Guo, H., L. Xu, A. Bougiatioti, K. M. Cerully, S. L. Capps, J. R. Hite, A. G. Carlton, S.-H. Lee, M. H. Bergin, N. L. Ng, A. Nenes, and R. J. Weber (2015), Predicting particle water and pH in the southeastern United States, Atmos. Chem. Phys., 15, 5211–5228. Jia, S., X. Wang, Q. Zhang, S. Sarkar, L. Wu, M. Huang, J. Zhang, and L. Yang Technical note: Comparison and interconversion of pH based on different standard states for aerosol acidity characterization, Atm. Chem. Phys., 18, 11125-11133. Nah, T., H. Guo, A. P. Sullivan, Y. Chen, D. J. Tanner, A. Nenes, A. Russell, N. L. Ng, L. G. Huey, and R. J. Weber (2018), Characterization of Aerosol Composition, Aerosol Acidity and Water-soluble Organic Acids at an Agriculture-intensive Rural Southeastern U.S. Site, Atm. Chem. Phys. , 18, 11471-11491.
* * *

---

## Referee Comment (RC3) · Anonymous Referee #3 · 21 Apr 2019

The paper used E-AIM model to calculate the pH of six Canadian cities over the course of 10 years. The paper states that (1) summer pH is about 1 unit lower than winter aerosol pH; (2) the pH is dependent more on meteorological conditions in the summer-time. In winter time both chemical composition and meteorological conditions influence the pH.

This paper is written clearly with a dataset that is suitable to be published in ACP. As the author stated, this paper is probably "the first long-term aerosol pH study in Canada" and provides the "longest records for evaluation of trends in the world". I enjoyed

reading the paper and suggest the authors consider the following before having the manuscript published.

The author made a conclusion on the effects of temperature on aerosol pH in page 6, line 20-22: "This result suggests the central role of meteorological conditions, especially temperature, in the determination of aerosol pH seasonal cycle in mid- and high-latitude regions. " Despite that the evidence raised by the author did support this argument, I think extending the 6 city dataset to all "mid- and high-latitude regions" is a bit too strong. For instance, what if there are farmland areas where ammonia emissions have a strong seasonal cycle. Then the temperature factor may not be the only dominant factor. I would advise the author to make some restrictions on this sentence.

One of the key assumptions in Figure 5 and 8(a) is that the sulfate:TNO3 ratio is 1:2. The author backed up this assumption in the text because in Toronto the average ratio of sulfate:TNO3 over the course of 10 years is about 1:2. However, since figure 5 examines the effects of chemical composition and temperature on pH from a seasonal perspective, is the sulfate:TNO3 ratio still 1:2 for each season? The author should show seasonal averaged information of sulfate:TNO3 to support that the ratio is still  $\sim$ 1:2, otherwise a sensitivity study of how sulfate:TNO3 influences the pH should be shown in the manuscript. Figure 5 should also simulate each season based on the actual sulfate:TNO3 ratios.

The authors stated that "Noticeably, when aerosol pH shows a decreasing trend with TNO3 concentration, there is still excess NH3 in the gas phase" in page 9, line 17. Theoretically, there will always be some level of NH3 in the gas phase, despite the absolute value might be small. Therefore, the author should define what "excess NH3" means here. Maybe a plot of NH3 concentration as a function of TNO3 concentration should be shown in Figure 8 as well to show that the NH3 concentration does not change too much with increasing TNO3 values. Maybe the authors could also add a couple of sentences to clarify why a decreasing pH would still lead to an excess NH3 concentration.

**ACPD**
Besides the typos mentioned by the previous reviewers, another typo I noticed is that ACPD

References:

the state "Vermont" is misspelled in Figure 1.

Weber, R. J., et al. (2016). "High Aerosol Acidity Despite Declining Atmospheric Sulfate Concentrations over the Past 15 Years." Nature Geosci 9(4): 282-285.

Guo, H., et al. (2015). "Fine-Particle Water and Ph in the Southeastern United States." Atmos. Chem. Phys. 15(9): 5211-5228.

---

## Referee Comment (RC4) · Anonymous Referee #4 · 13 May 2019

Overall, the paper is well written and contributes meaningful analysis to the community. The authors could further justify their approach (general comment 1) and provide more supporting information (specific comment 2).

General comments: 1. Given the importance of meteorological drivers of pH (RH and T) indicated by the analysis, how did the authors justify the use of daily average RH, T, and composition led to the appropriate average pH? How important are diurnal variations in RH, T, and composition in dictating daily average pH?

[Figure]

2. The manuscript presents evidence that long-term changes in pH are driven by changes in RH and T (Fig 6-7), however, sulfate likely correlates with T. St. Anicet and Toronto do not have statistically significant changes in sulfate over the time period. Is pH at those sites related to sulfate?

Specific comments:

1. In a couple place, the authors mention "ion balance" which could imply they used a charge balance. I suggest rewording on page 1, line 9 and page 4, line 21. Page 4, line 21 indicates SNA contributed more than 80% of total charges. What is the remaining 20%? Salts (NaCl) and NVC are neglected. Are they important?

2. At a minimum, tabulated pH values should be provided in the SI. The authors should consider providing additional data and/or model inputs/outputs for data documentation purposes.

---

## Author Comment (AC1) · 17 Jun 2019

Reviewer 1#

General Comments:

This paper presents aerosol pH predictions at six Canadian sites based on long-term aerosol composition and NH3 monitoring data. The authors characterize temporal trends in aerosol pH and seasonal averages at each site. Using the observational data, the authors characterize the sensitivity of pH to meteorology (Temp and RH) and composition (sulfate-nitrate-ammonium). The authors find that pH has a distinct seasonal profile at all of the sites, with summertime pH systematically lower than wintertime pH. During summer, aerosol pH is much less sensitive to perturbations in composition than it is to the met parameters. During winter, aerosol pH is still quite sensitive to meteorology, but is more sensitive to compositional changes than during summer. Overall, these results are novel and likely to be of interest to many in the atmospheric chemistry community. The topic is certainly relevant for ACP. The writing is clear, and the conclusions are robustly supported by the measurements and modeling results. I strongly endorse the manuscript for publication after the following specific concerns are addressed.

Specific Comments:

It is important that the authors state that their findings only apply to the sulfate-nitrate-ammonium system. Locations with dust and/or marine influence may exhibit very different trends and sensitivities to meteorology and composition.
[Thanks. We have included this statement in the conclusion part.]

"…This study focused on a number of sites with relatively low ambient mass loadings of aerosol inorganic constituents dominated by sulfate-nitrate-ammonium system. Regions in which $PM_{2.5}$ is strongly influenced by dust or marine sources may have different sensitivities for chemical and meteorological factors. However, for places…"

The authors should discuss possible sampling artifacts as a function of temperature and season. The discussion on Pg. 4, lines 1-2 is presented in a way that assumes no sampling artifact, but studies (including the Yu et al. (2006) study that the authors cited) show evaporation of NH4NO3 from nylon filters. There is a temperature and RH-dependence to these losses, which are often greatest in summer. How would even small sampling artifacts during summer influence these results?
[In this sampling apparatus, a backup nylon filter is used to capture the volatilized nitrate from the Teflon filter and to correct for semi-volatile ammonium nitrate loss. The subsequent evaporation of nitrate from the nylon filter has been proved to be negligible by Yu et al. (2006).]

How were solids treated (prohibited or allowed?) in the model runs? How were data points with very low RH and ALW treated? Presumably, there were more of these points during winter, when dry, artic air influences many of the sites? The interpretation of pH under conditions with a low fraction of ALW seems questionable.
[In our pH calculation, aerosol is forced to be metastable with no solid formation allowed, as mentioned in section 2.4. Our data show that almost all sampling days had average RH higher than 40%, larger than the efflorescence relative humidity of ammonium sulfate, so the aerosol is likely to have deliquesced during the diurnal variation of RH and T. After selecting all the sampling events with RH<40%, as shown in the below figure, the predicted F(NH₃) still has good agreement with the

measured F(NH₃) for most of the data points. As a result, this assumption does not have significant impact on our calculation.]

[Figure]

The authors should replicate Figure S2 with the predicted and measured ε(NO3).

[We put the comparison between predicted and measured $\varepsilon(NO_3)$ in the graph below, which showed poor correlation. According to the results shown in Figure S5, for most of the model output, nitrate phase partitioning tended toward extreme values ($\varepsilon(NO_3)$ <20 % or >80 %). In contrast, measured $\varepsilon(NO_3)$ are distributed more evenly. Combined with the low concentration levels at these sites, both the measurement uncertainties and the presence of non-volatile cations drive large uncertainties in $\varepsilon(NO_3)$. We also have evidence to suggest that sometimes it's improper to use daily average concentration and meteorological parameters to calculate average $\varepsilon(NO_3)$, and it is also questionable to involve all non-volatile cations in predicting nitrate phase partitioning. We intend to address this issue more fully in a separate study.]

[Figure]

While the authors find only minor effects of non-volatile cations (NVC) in their analysis, the limitations of this analysis need to be discussed. For example, Ca and Mg salts are typically much less soluble than analogous Na salts – replacing Ca and Mg with equivalent Na therefore does not account for potential precipitation, which may have a more pronounced effect on pH. Na is also more hygroscopic, which will affect the predicted ALW. Also, a brief discussion on the effects of NVC on pH (see Fig. 4c of Guo et al., 2018) should be included in the intro where composition effects on pH are discussed.

[We have now added more discussion about how this assumption will affect the prediction results in this paragraph.]

"Using $Na^+$ to replace other non-volatile cations does not account for the precipitation of $CaSO_4$, which has the same effect on aerosol pH of reducing sulfate. The substitution also does not perfectly reflect the impact on aerosol liquid water content, which may indirectly affect the pH. In general, the gas fraction of NHx calculated from the output of the thermodynamic model matches very closely with the measured gas fraction of NHx (Figure S2), suggesting that we are not missing substantial contributions to the ion balance in the particles by only considering ammonium sulfate."

Temperature and RH are strongly linked in the atmosphere: discussing them as if they are independent parameters is not correct. A missing element from this manuscript is the role and trends in ALW, which is also closely connected to T and RH. Nguyen et al. (2016) show strong seasonal trends in ALW in the Northeast USA (see Fig. 2d), which would presumably have similar behavior to the Canadian sites analyzed in the present manuscript. If the Canadian sites do indeed show similar behavior (ALW maxima in summer), then it would imply the more acidic summertime particles occur despite the diluting effect of the ALW enhancement. This would be a very interesting contrast to the results of Guo et al. (2015) and Battaglia et al. (2017), who both show that aerosol pH decreases with increasing T. In their studies, the increase in T (and decrease in RH) that happen in the afternoon lead to decreasing ALW and a concentrating effect on H+ (hence, decreasing pH). Clearly, the seasonal changes in

composition would be an important factor that also contributes to these observations.

[The plot of daily average RH and T shows very poor correlation (as shown in the figure below for the Toronto site, the other five sites all have similar pattern), suggesting that we do not need to be concerned about this as a confounding factor in our interpretation of the dominant cause of the seasonal variation pattern of aerosol pH. Based on our conceptual modelling of aerosol pH sensitivity to RH and T in Figure 4, it can be seen that under fixed chemical composition and temperature, the effect of changing relative humidity in changing pH is relatively small compared to temperature. Changing relative humidity from 40% to 90% generally can change aerosol pH within 0.5 unit, while the seasonal variation of temperature can drive aerosol pH oscillating between 2 and 4.]

[Figure]

I found the explanation of Fig. 8b (pg. 9, line 12-24) to be quite confusing.

[We have rewritten this part to describe the process in more detail.]

"…However, the effect of the addition of total nitrate is more complicated. Based on equation [2], the added $TNO_3$ concentration can impact aerosol pH in two opposite ways. First, because ammonium nitrate is more hygroscopic than ammonium sulfate (Gysel et al., 2007), the particulate nitrate formation will raise the liquid water content [ALW] in aerosol, increasing the aerosol pH; however, nitrate formation will also scavenge $NH_3$ from gas phase, and the smaller value of $[NH_3]/[NH_4^+]$ will make aerosol more acidic. These two factors altogether contribute to the arc-shaped curve of the relationship between aerosol pH and nitrate shown in Figure 8(b). Figure S7 shows…"

I completely disagree with the first sentence in the Abstract. Even if the chemical composition of an aerosol is known precisely, it is not possible to calculate the pH from an ion balance due to the buffering

effects the HSO4-/SO42- equilibrium (similar effect for organic acids, as well). Further in the manuscript – pg. 2 line 12 – the authors cite several studies which also contradict this statement. To my knowledge, there has never been a study which demonstrated a good connection between the ion balance and pH, so I think it is a mistake to lead the manuscript with this idea.
[Thanks for pointing that out. We have changed this description.]

"…calculate the aerosol water content and the $H^+$ concentration through the equilibrium among acids and their conjugate bases."

Technical Corrections:
Pg. 1, line 10 "constitute" or "constituent"?
[We have made the change.]

Pg. 2, line 7: say "aerosol aqueous phase" instead of "aerosol liquid water"
[We have made the change.]

Pg. 2, line 8: suggest deleting "very"
[We have made the change.]

Pg. 6, line 26-28: this seems important - explain the derivation of this theoretical relation. Also, can the authors compare the prediction to their data?
[We have extended this part for detailed explanation.]

"…Under equilibrium, aerosol pH can be theoretically derived from the observed gas/particle concentration ratio of NHx and aerosol liquid water [ALW] as: $pH = \log([NH_3]/[NH_4^+]) + \log[ALW] + pKa + \log K_H$, which Hennigan et al. (2015) showed having good agreement with E-AIM modelling results. Because both $K_H$ and pKa have strong temperature dependencies (Chameides, 1984; Bell et al., 2007), aerosol pH is going to be temperature-dependent even if liquid water content or $NH_3/NH_4^+$ partitioning behavior does not change. The partial derivative of aerosol pH dependence on temperature will give $\partial pH/\partial T = \partial(pKa + \log K_H)/\partial T \approx -0.05$ $(K^{-1})$, which corresponds to 0.1 unit increase (decrease) of aerosol pH if temperature decreases (increases) by 2 $^{\circ}C$. The pH gradient shown in Figure 4 also illustrated that approximately 10 $^{\circ}C$ increase of temperature is required for 0.5 unit decrease in aerosol pH under the same chemical composition and RH, which is consistent with the temperature sensitivity derived through NHx phase partitioning method."

Pg. 7, line 29 (pg. 9, line 14): I suggest alternate wording ("aerosol was more neutralized") – because the pH is still quite acidic, suggest using "aerosol pH was higher" or similar.
[We have made the change.]

Pg. 10, line 6-7; why would aerosol loading affect the results?
[We will rephrase this sentence with more accurate description. According to the aerosol pH calculated under different chemical composition shown in Figure 5, it's both the concentration of particulate chemical composition and gas phase $NH_3$ that determine the aerosol pH sensitivity.]

"…for places with high particulate pollution and different concentration level of NH₃, similar approaches can also be applied…"

The role of organosulfates (mentioned pg. 8, line 5 and pg. 9, line 23) in aerosol pH is not clear? Can the authors add clarification?
[We intend to make a point that there is a possibility of misrepresenting organosulfates as inorganic sulfate. We have added this explanation in the main text]

"…the potential contribution from organic acids or organosulfates (the error caused by misrepresenting organosulfates as inorganic sulfate) will also have a more significant impact…"

The color scale on Figures 4, 5, and 8 can be difficult to distinguish certain levels – the color scale used with Fig. 2 is much easier to discern.
[We have refined the color used for plotting these figures.]

Address grammar and/or awkward phrasing in the following places: pg. 6, line 26; pg. 7 line 28; pg. 7 line 30.
[We have rephrased these sentences.]

"…aerosol pH is going to be temperature-dependent even if…"
"…where it shows that with fixed chemical composition, aerosol will be slightly less acidic at higher RH."
"…Apart from the effect of lower temperature, lower concentration of NHx and lower NHx to sulfate molar ratio also made aerosol pH much more sensitive to chemical composition changes…"

References:
Battaglia, M. A., Douglas, S., and Hennigan, C. J.: Effect of the Urban Heat Island on Aerosol pH, Environmental Science & Technology, 51, 13095-13103, 10.1021/acs.est.7b02786, 2017.
Guo, H., Xu, L., Bougiatioti, A., Cerully, K. M., Capps, S. L., Hite, J. R., Carlton, A. G., Lee, S. H., Bergin, M. H., Ng, N. L., Nenes, A., and Weber, R. J.: Fine-particle water and pH in the southeastern United States, Atmos Chem Phys, 15, 5211-5228, 2015.
Guo, H., Nenes, A., and Weber, R. J.: The underappreciated role of nonvolatile cations in aerosol ammonium-sulfate molar ratios, Atmos. Chem. Phys., 18, 17307-17323, 2018.
Nguyen, T. K. V., V. P. Ghate, and A. G. Carlton (2016), Reconciling satellite aerosol optical thickness and surface fine particle mass through aerosol liquid water, Geophys. Res. Lett., 43, doi:10.1002/2016GL070994.
Yu, X. Y., Lee, T., Ayres, B., Kreidenweis, S. M., Malm, W., and Collett, J. L.: Loss of fine particle ammonium from denuded nylon filters, Atmospheric Environment, 40, 4797-4807, 10.1016/j.atmosenv.2006.03.061, 2006.

---

## Author Comment (AC2) · 17 Jun 2019

**Reviewer 2**

This paper determines particle pH over a 10-year period at a number of sites in eastern Canada. pH is reported to be higher in winter vs. summer by about one pH unit, similar to what has been reported in Atlanta, although the Atlanta aerosol pH is a bit lower in both seasons [Guo et al., 2015]. The authors conclude that the difference in pH is largely due to T. Another major finding is that pH is reported to be more variable in winter due to particle composition effects. The topic is appropriate for publication in ACP. The paper is an important contribution to the growing knowledge on fine particle pH. There are a number of points the authors could consider.

1) Although it is concluded that T is the main driver of lower pH in summer, with minor influence by aerosol composition, ambient data did not seem to be used in the assessment. For example, maybe a comparison table could be made showing summer vs winter average concentrations of the main species involved (sulfate, ammonium and nitrate). (I would also suggest including the ratio of nitrate to sulfate, which may be important for the next point.) Note that Guo et al (2015) reported a role of higher ion concentrations (sulfate) in summer as a cause for their reported differences in summer and winter pHs, but did not do a detailed investigation specifically on the effect of temperature. This paper can address this issue in much more detail and provide a more definitive conclusion. As noted in this paper, some of the T effects result from the equilibrium constants (Henry's law and dissociation, Ka's) depending on T. Guo et al (e.g., see [Guo et al., 2017] supplemental material: https://www.atmos-chem-phys.net/17/5703/2017/acp-17-5703-2017-supplement.pdf) has shown one way to look at T (and particle water effects) is through S curves of partitioning vs pH. I think this is a useful way to think of things and it may be useful here. Overall, I think more insight could be gained if the authors tried to remove or account for the T effects in their data and then look for other possible drivers of pH differences.

[In this paper, we intend to mainly discuss the aerosol pH response to each factor by interpreting how these factors can influence the equilibrium:  $NH_3(g) \leftrightarrow NH_3(aq)$  and  $NH_4^+ \leftrightarrow NH_3(aq) + H^+$ . The theoretical derivative of aerosol pH from the above two equilibria gives:  $pH = \log([NH_3]/[NH_4^+])+\log[ALW]+pKa+\logK_H$ . The S curve of partitioning vs pH as mentioned by the reviewer mainly discuss how pH will respond to  $[NH_3]/[NH_4^+]$ , while our paper mainly talks about the influence on pKa and KH. We mainly focus on the ten-year time series of pH and try to identify the major influencing factors for the seasonal oscillation pattern. One piece of strong evidence for the dominant influence of temperature on pH is Figure 4 showing that the seasonal pattern of aerosol pH variation can be reconstructed without changing chemical composition. We can also theoretically derive that about 26% increase in NH3 concentration is required to off-set the effect of temperature decreasing 2 °C if RH and particle composition do not change. We agree that the change in chemical composition can also affect or even dominate pH variation if the concentration change is large enough. However, we do not intend to have too much quantitative discussion in this paper but mainly try to identify the most important factor.]

2) The contrast in particle chemical composition effect on pH could be explored further by investigating the role of particle nitrate, or with nitrate to sulfate ratios. The paper tends to hint at the role of nitrate on this effect, but maybe it is playing a very large role here. If so, the paper could be more precise than just stating composition effects. For example, in summer the nitrate levels are likely to be low because of the higher temperatures favoring the gas phase, resulting in low NO3-/SO42-

ratios. Particle water, in the summer, is then largely controlled by the main hygroscopic species, nonvolatile sulfate (and T, RH). pH depends on the inverse of particle water concentration. In contrast, in winter, the NO3-/SO42- ratio is likely much higher since nitrate partitioning to the particle is more favored. (I.e., at lower T, there is a higher predicted fraction of NO3- to TNO3 at a given pH; see LWC and T affect on nitrate partitioning vs pH S curves in Guo et al. (2017) supplemental material). Indeed, Fig S5 shows that based on this data set, a small fraction of nitrate partitions to the aerosol in summer, whereas a much larger fraction is in the particle phase in winter. This means that in winter liquid water is affected by both sulfate and nitrate, and possibly more so by nitrate. Nitrate is semi-volatile, which leads to feed backs [Guo et al., 2017]. The last figure shows part of this dependence; initially the uptake of nitrate leads to higher liquid water (nitrate is highly hygroscopic), which dilutes the H+ and raises pH. If nitrate partitioning is in the sensitive part of the S curve then higher pH leads to more nitrate, which allows for more uptake of nitrate (see again S curves for nitrate partitioning and LWC effects). Hence the more partitioning of nitrate drives the pH higher (as seen in Fig 8b, although this is overwhelmed at some point apparently by the increase in H+ from the dissociation of HNO3 aq). Also, with high particle nitrate the system becomes more sensitive to T, and so overall it might simply be the higher nitrate levels in winter cause greater variability in winter pH.

[We agree that the difference in nitrate to sulfate molar ratio can also affect the aerosol pH sensitivity to chemical composition, so we will put more relevant statement when discussing chemical composition effect. However, we also wish to point out that we found the impact of nitrate was limited in the regions we studied, and even in wintertime it is both lower NH3 concentration and more nitrate that jointly lead to the greater pH variability. Higher nitrate in winter alone cannot explain the whole pattern, and this can also be seen from the site by site comparison that the sites with lower NHx to sulfate molar ratio (as an indicator for lower NH3) tended to have larger sensitivity to composition changes.]

"...During summer ammonium nitrate formation is unfavorable, while in wintertime ammonium nitrate can form very efficiently. As a result, in summertime, the aerosol liquid water content was mainly contributed by ammonium sulfate while in wintertime it was affected by both sulfate and nitrate. The effect of nitrate formation to aerosol pH will be further discussed in section 3.4."

"...Apart from the effect of lower temperature, lower NHx to sulfate molar ratio also made aerosol pH much more sensitive to chemical composition changes than the other seasons because it will make the  $log([NH_3]/[NH_4^+])$  part in equation 2 more sensitive to chemical component changes..."

"...However, the effect of the addition of total nitrate is more complicated. Based on equation [2], the added TNO3 concentration can impact aerosol pH in two opposite ways. First, because ammonium nitrate is more hygroscopic than ammonium sulfate (Gysel et al., 2007), the particulate nitrate formation will raise the liquid water content [ALW] in aerosol, increasing the aerosol pH; however, nitrate formation will also scavenge NH3 from gas phase, and the smaller value of [NH3]/[NH4+] will make aerosol more acidic. These two factors altogether contribute to the arc-shaped curve of the relationship between aerosol pH and nitrate shown in Figure 8(b). Figure S7 shows..."

**Minor Comments.**

First line of Abstract that states: Aerosol pH is difficult to measure directly but can be calculated if the

chemical composition is known with sufficient accuracy and precision to calculate the aerosol water content and the H+ concentration through ion balance. Strictly speaking this is true, but in practical terms can it be done. For example, there is the issue of measurement uncertainty and propagation of those errors, leading to high uncertainty in H+. One must also measure all ions at actual ambient aerosol liquid water concentrations, eg, one would have to measure bisulfate and sulfate levels that exist in the ambient aerosol accurately. Given that ion balances have been used in the past to incorrectly infer acidity, I suggest removing this sentence to avoid perpetuating this idea? [We agree with this point and have changed this statement.]

"...calculate the aerosol water content and the H+ concentration through the equilibrium among acids

and their conjugate bases."

Second line of Abstract that states: In practical terms, simultaneous measurements of at least one semivolatile constitute, e.g. NH3 or HNO3, are required to provide a constraint on the calculation of pH. Is this strictly true as a general rule? Consider the sigmoid curves for the fractional partitioning of a semivolatile species vs pH (see Guo et al. 2017 supplemental material). For a given set of conditions, there is only about a 1.5 to 2 pH range that partitioning is sensitive to pH, outside this range there is very little sensitivity (e.g., either all is in gas or all in particle phases). This line in the text could be clarified by stating that the semi-volatile constitute used to constrain the pH calculations should be in both the particle and gas phase, e.g., epsilon (=p/(g+p) or g/(g+p) ) between say 20 and 80%. It then raises the question of how one actually goes about solving for a final pH given missing species (i.e., does one iterate, is for how many times...).

[We agree that for the S curve there is a range of pH corresponding to sharp change in the phase partitioning of semi-volatile bases or acids. However, we'd like to make a point that the pH being insensitive to phase partitioning is not equivalent to pH being poorly constrained or less reliably calculated. Even gas phase (or particle phase) fraction does not fall into 20% to 80%, pH can still be reliably calculated as long as the measurement has enough accuracy. For example, if Frac(NH3)>80%, aerosol pH can still be reliably calculated with the accurate input, while the difference is that pH is less sensitive to NH3 concentration under higher Frac(NH3) conditions.]

Line 15, as noted above, measured NH3/NH4+ constraint on pH only works when there is some fraction in each phase. If the particle pH is very low or high, then it is possible for the system of have practically no gas phase NH3 present or no particle phase NH4+ present, in which case the observed concentrations of these species do not provide a good constraint on pH.

[We agree that there is an optimal range for  $NH_3/NH_4^+$  constrained pH calculation. However, we wish to note that there is no clear boundary between whether it is  $NH_3/NH_4^+$  affecting aerosol pH, or aerosol pH affecting  $NH_3/NH_4^+$  phase partitioning. Especially for particles made mostly of ammonium salts, the equilibrium between  $NH_4^+$  and  $H^+$  is the strong constraint of aerosol acidity. Another way to look at it is through the interpretation of Figure 5, the largest aerosol pH gradient occurs in the regions where there is little free gas phase  $NH_3$ . In these regions, a small change in gas phase  $NH_3$  concentration will result in a large aerosol pH change. In this case, it is hard to tell whether it is the modelling that has the poor constraint or the aerosol pH itself actually dramatically changing.]

Section 2.4: Some things to consider briefly discussing regarding the model and its assumptions: Lack

of consideration of other species by E-AIM (e.g., non-volatile cations, which is discussed later in the paper), specific RH range data in this study (can one believe the predictions down to 30% and greater than 90% RH – why not test with comparison of observed/measured NH3 partitioning?), no consideration of organic species influence on LWC, and no consideration of possible phase separations (ie, more important at lower RH). Some of these issues are noted at the very end of the paper, it might be worth point them out when the thermodynamic model is discussed.

[We have also now included these limitations in the discussion of aerosol pH calculation.]

"... The reliability of the pH calculations depends on several assumptions, including that daily average values are appropriate for the calculations, and that the gas and particles phases are equilibrated. The limitations of using E-AIM II model also include the lack of consideration of non-volatile cations, aerosol liquid water contributed by organic species or possible phase separation etc. One rigorous method to evaluate the reliability of the calculated pH is to compare the input (measured) and output (modelled) gas-particle partitioning of semi-volatile species..."

This paper often refers to an earlier paper by Murphy et al. 2017 Faraday Disc. In that paper the pH was calculated differently; there is a difference of a factor of 55.509, the conversion from mole-based to molality-based activity. Does this account for the generally higher pH reported in Murphy et al of 2.5 to 5.5 for a similar region as discussed in this paper (ie, that paper reported molarity based pH, whereas this paper molality based pH)? Also see [Jia et al.]. I think the difference in pH calculations (and results) between these two papers should be noted.

[In the final published version of Murphy et al. 2017 Faraday Disc paper, the pH is calculated with the molality-based activity, which is consistent with our current paper. The values of aerosol pH calculated in the two papers fall within the approximately same range.]

Page 5 and Fig S2. The comparisons between measured and predicted NH3 and NH4+ are very useful, but why not show a similar set of plots in the supplementary for HNO3 and NO3-. This would provide a more complete assessment of the model predictions and a useful contrast to what has been reported in a number of other papers which show that ammonia/ammonium partitioning is much better predicted by thermodynamic models than nitrate partitioning [Guo et al., 2017; Guo et al., 2016], [Guo et al., 2018] supplemental material, and [Nah et al., 2018] supplemental material.

[We put the comparison between predicted and measured  $\epsilon(NO_3)$  in the graph below, which showed poor correlation. According to the results shown in Figure S5, for most of the model output, nitrate phase partitioning tended toward extreme values ( $\epsilon(NO_3) < 20 \%$  or >80 %). In contrast, measured  $\epsilon(NO_3)$  are distributed more evenly. Combined with the low concentration levels at these sites, both the measurement uncertainties and the presence of non-volatile cations drive large uncertainties in  $\epsilon(NO_3)$ . We also have evidence to suggest that sometimes it's improper to use daily average concentration and meteorological parameters to calculate average  $\epsilon(NO_3)$ , and it is also questionable to involve all non-volatile cations in predicting nitrate phase partitioning. We intend to address this issue more fully in a separate study.]

---

## Author Comment (AC3) · 17 Jun 2019

Reviewer 3#

The paper used E-AIM model to calculate the pH of six Canadian cities over the course of 10 years. The paper states that (1) summer pH is about 1 unit lower than winter aerosol pH; (2) the pH is dependent more on meteorological conditions in the summertime. In winter time both chemical composition and meteorological conditions influence the pH.

This paper is written clearly with a dataset that is suitable to be published in ACP. As the author stated, this paper is probably "the first long-term aerosol pH study in Canada" and provides the "longest records for evaluation of trends in the world". I enjoyed reading the paper and suggest the authors consider the following before having the manuscript published.

The author made a conclusion on the effects of temperature on aerosol pH in page 6, line 20-22: "This result suggests the central role of meteorological conditions, especially temperature, in the determination of aerosol pH seasonal cycle in mid- and high-latitude regions. "Despite that the evidence raised by the author did support this argument, I think extending the 6 cities dataset to all "mid- and high-latitude regions" is a bit too strong. For instance, what if there are farmland areas where ammonia emissions have a strong seasonal cycle. Then the temperature factor may not be the only dominant factor. I would advise the author to make some restrictions on this sentence.

[We agree to this point. Our conclusion stands when the chemical composition has NHx>> sulfate, as it will be discussed in the later section that lower NHx:sulfate molar ratio will lead to larger sensitivity to composition changes, so we have revised this sentence to a more rigorous description.]

"This result suggests the central role of meteorological conditions, especially temperature, in the determination of aerosol pH seasonal cycle in mid- and high-latitude regions with NHx >> sulfate in chemical composition."

One of the key assumptions in Figure 5 and 8(a) is that the sulfate:TNO3 ratio is 1:2. The author backed up this assumption in the text because in Toronto the average ratio of sulfate:TNO3 over the course of 10 years is about 1:2. However, since figure 5 examines the effects of chemical composition and temperature on pH from a seasonal perspective, is the sulfate:TNO3 ratio still 1:2 for each season? The author should show seasonal averaged information of sulfate:TNO3 to support that the ratio is still  $\sim$ 1:2, otherwise a sensitivity study of how sulfate:TNO3 influences the pH should be shown in the manuscript. Figure 5 should also simulate each season based on the actual sulfate:TNO3 ratios. [We added the calculation showing the influence of this assumption to aerosol pH in each month in each sampling site.]

"One key assumption in the above conceptual modelling is forcing total nitrate to sulfate molar ratio to be 2. To test the effect of this assumption on the aerosol pH, we calculate the pH of aerosol on each sampling day assuming total nitrate to sulfate molar ratio to be 2 and compare with the pH calculated with the measured total nitrate. The results show that this assumption has negligible influence on aerosol pH for samples from April to November but has larger influence on wintertime aerosol, indicating that wintertime aerosol is more sensitive to nitrate concentration (the statistical summary is shown in Figure S6). The influence of nitrate to wintertime aerosol pH will be further discussed in section 3.4."

The authors stated that "Noticeably, when aerosol pH shows a decreasing trend with TNO3 concentration, there is still excess NH3 in the gas phase" in page 9, line 17. Theoretically, there will always be some level of NH3 in the gas phase, despite the absolute value might be small. Therefore, the author should define what "excess NH3" means here. Maybe a plot of NH3 concentration as a function of TNO3 concentration should be shown in Figure 8 as well to show that the NH3 concentration does not change too much with increasing TNO3 values. Maybe the authors could also add a couple of sentences to clarify why a decreasing pH would still lead to an excess NH3 concentration.

[We have rewritten this part to focus on the effect of ammonium nitrate formation on aerosol pH.]

"...However, the effect of the addition of total nitrate is more complicated. Based on equation [2], the added TNO3 concentration can impact aerosol pH in two opposite ways. First, because ammonium nitrate is more hygroscopic than ammonium sulfate (Gysel et al., 2007), the particulate nitrate formation will raise the liquid water content [ALW] in aerosol, increasing the aerosol pH; however, nitrate formation will also scavenge NH3 from gas phase, and the smaller value of  $[NH_3]/[NH_4^+]$  will make aerosol more acidic. These two factors altogether contribute to the arc-shaped curve of the relationship between aerosol pH and nitrate shown in Figure 8(b). Figure S7 shows..."

Besides the typos mentioned by the previous reviewers, another typo I noticed is that the state "Vermont" is misspelled in Figure 1.

[Thanks for pointing out. We have corrected this typo.]

**References:**

Weber, R. J., et al. (2016). "High Aerosol Acidity Despite Declining Atmospheric Sulfate Concentrations over the Past 15 Years." Nature Geosci 9(4): 282-285.

Guo, H., et al. (2015). "Fine-Particle Water and Ph in the Southeastern United States." Atmos. Chem. Phys. 15(9): 5211-5228.

---

## Author Comment (AC4) · 17 Jun 2019

**Reviewer 4**

Overall, the paper is well written and contributes meaningful analysis to the community. The authors could further justify their approach (general comment 1) and provide more supporting information (specific comment 2).

General comments: 1. Given the importance of meteorological drivers of pH (RH and T) indicated by the analysis, how did the authors justify the use of daily average RH, T, and composition led to the appropriate average pH? How important are diurnal variations in RH, T, and composition in dictating daily average pH?

[This is an important question given that many monitoring datasets available for aerosol pH calculation provide only 24-hour observation of chemical composition. The diurnal changes of RH and T can also drive diurnal variation in aerosol pH, just as they do on a seasonal scale. If both chemical composition and meteorological factors are changing within the 24-hour sampling period, the definition of the "average pH" of these particles are crucial for the justification of the calculation. Here we simply define the average pH of aerosol as pH calculated under 24-hour average concentrations and meteorological parameters, and evaluate how well this can successfully reconstruct the phase partitioning of NHx. The degree to which the diurnal variation of pH can deviate from this average value cannot be determined with the 24-hour average data, but is related to the sensitivity to each factor, which is partially addressed by this paper. For example, we have shown that  $\partial pH/\partial T\approx -0.05$  (K-1), so this rule will also hold for the diurnal pH variation. The graph below shows the comparison of summertime Toronto aerosol pH calculated with the 24-hour average of the hourly pH values ( the data are from VandenBoer et al. (2011)), which showed high consistency. This result indicates that using daily average parameters to calculate aerosol pH is reliable.]

2. The manuscript presents evidence that long-term changes in pH are driven by changes in RH and T (Fig 6-7), however, sulfate likely correlates with T. St. Anicet and Toronto do not have statistically significant changes in sulfate over the time period. Is pH at those sites related to sulfate?

[On Figure 6 and 7, if we select data of Toronto and St Anicet sites from year 2013 to 2016, we can see that sulfate concentrations were continuously decreasing but aerosol pH at these two sites both showed minimum values in year 2014 due to the highest temperature at that year. This pattern indicates that aerosol pH was more significantly affected by temperature than by sulfate.]

Specific comments:

1. In a couple place, the authors mention "ion balance" which could imply they used a charge balance. I suggest rewording on page 1, line 9 and page 4, line 21. Page 4, line 21 indicates SNA contributed more than 80% of total charges. What is the remaining 20%? Salts (NaCl) and NVC are neglected. Are they important?

[We have reworded these two sentences. We also have assessed the influence of NVC by comparing the pH calculated with E-AIM II and E-AIM IV model, suggesting that the NVC will not affect the pH significantly. We also have added more discussion about this result in the response to Reviewer 2# comment.]

"...the  $H^+$  concentration through the equilibrium among acids and their conjugate bases..."

"...more than 80% of total measured charges in the particles..."

"Using Na+ to replace other non-volatile cations does not account for the precipitation of CaSO4, which has the same effect on aerosol pH of reducing sulfate. The substitution also does not perfectly reflect the impact on aerosol liquid water content, which may indirectly affect the pH. In general, the gas fraction of NHx calculated from the output of the thermodynamic model matches very closely with the measured gas fraction of NHx (Figure S2), suggesting that we are not missing substantial contributions to the ion balance in the particles by only considering ammonium sulfate."

2. At a minimum, tabulated pH values should be provided in the SI. The authors should consider providing additional data and/or model inputs/outputs for data documentation purposes.

[We will upload our dataset about chemical composition and pH calculation in the supplement excel file of this paper.]

VandenBoer, T. C., Petroff, A., Markovic, M. Z., and Murphy, J. G.: Size distribution of alkyl amines in continental particulate matter and their online detection in the gas and particle phase, Atmospheric Chemistry and Physics, 11, 4319-4332, 10.5194/acp-11-4319-2011, 2011.